# Genomic Analysis of Two MDR Isolates of *Salmonella enterica* Serovar Infantis from a Spanish Hospital Bearing the *bla*_CTX-M-65_ Gene with or without *fosA3* in pESI-like Plasmids

**DOI:** 10.3390/antibiotics11060786

**Published:** 2022-06-09

**Authors:** Xenia Vázquez, Javier Fernández, Jesús Rodríguez-Lozano, Jorge Calvo, Rosaura Rodicio, M. Rosario Rodicio

**Affiliations:** 1Área de Microbiología, Departamento de Biología Funcional, Universidad de Oviedo (UO), 33006 Oviedo, Spain; xenia_grao@hotmail.com; 2Instituto de Investigación Sanitaria del Principado de Asturias (ISPA), 33011 Oviedo, Spain; javifdom@gmail.com (J.F.); mrosaura@uniovi.es (R.R.); 3Servicio de Microbiología, Hospital Universitario Central de Asturias (HUCA), 33011 Oviedo, Spain; 4Research & Innovation, Artificial Intelligence and Statistical Department, Pragmatech AI Solutions, 33003 Oviedo, Spain; 5Centro de Investigación Biomédica en Red-Enfermedades Respiratorias, 20029 Madrid, Spain; 6Servicio de Microbiología, Hospital Universitario Marqués de Valdecilla (IDIVAL), 39008 Santander, Spain; jesus.rodriguez@scsalud.es (J.R.-L.); jorge.calvo@scsalud.es (J.C.); 7CIBER de Enfermedades Infecciosas, Instituto de Salud Carlos III, 28029 Madrid, Spain; 8Departamento de Bioquímica y Biología Molecular, Universidad de Oviedo (UO), 33006 Oviedo, Spain

**Keywords:** *S*. Infantis, multidrug-resistance, *bla*
_CTX-M-65_, *fosA3*, heavy-metal resistance, pESI-like megaplasmid, whole-genome sequencing

## Abstract

*Salmonella enterica* serovar Infantis (*S*. Infantis) is a broiler-associated pathogen which ranks in the fourth position as a cause of human salmonellosis in the European Union. Here, we report a comparative genomic analysis of two clinical *S*. Infantis isolates recovered in Spain from children who just returned from Peru. The isolates were selected on the basis of resistance to cefotaxime, one of the antibiotics of choice for treatment of *S. enterica* infections. Antimicrobial susceptibility testing demonstrated that they were resistant to eight classes of antimicrobial agents: penicillins, cephalosporins, phenicols, aminoglycosides, tetracyclines, inhibitors of folate synthesis, (fluoro)quinolones and nitrofurans, and one of them was also resistant to fosfomycin. As shown by whole-genome sequence analysis, each isolate carried a pESI-like megaplasmid of ca. 300 kb harboring multiple resistance genes [*bla*_CTX-M-65_, *aph(4)-Ia*, *aac(3)-IVa*, *aph(3′)-Ia*, *floR*, *dfrA14*, *sul1*, *tet*(A), *aadA1* ± *fosA3*], as well as genes for resistance to heavy metals and disinfectants (*mer*, *ars* and *qacEΔ1*). These genes were distributed in two complex regions, separated by DNA belonging to the plasmid backbone, and associated with a wealth of transposable elements. The two isolates had a D87Y amino acid substitution in the GyrA protein, and truncated variants of the nitroreductase genes *nfsA* and *nsfB*, accounting for chromosomally encoded resistances to nalidixic acid and nitrofurantoin, respectively. The two *S*. Infantis isolates were assigned to sequence type ST32 by in silico multilocus sequence typing (MLST). Phylogenetic analysis revealed that they were closely related, differing only by 12 SNPs, although they were recovered from different children two years apart. They were also genetically similar to *bla*_CTX-M-65_-positive ± *fosA3* isolates obtained from humans and along the poultry production chain in the USA, South America, as well as from humans in several European countries, usually associated with a travel history to America. However, this is the first time that the *S*. Infantis *bla*_CTX-M-65_ ± *fosA3* MDR clone has been reported in Spain.

## 1. Introduction

*Salmonella enterica* serovar Infantis (*S*. Infantis) is one of the main non-typhoid serovars of *S. enterica*. In 2020, it ranked in the fourth position as a cause of human salmonellosis in the European Union, only preceded by *S*. Enteritidis, *S*. Typhimurium and the monophasic 1,4,[5],12:i:- variant of the latter serovar. In addition, it was the most common serovar in broiler flocks and broiler meat [1]. Apart from the European Union, *S*. Infantis has increasingly been found in clinical, animal and food samples from many other geographical regions [2,3,4,5,6]. The epidemiological success of *S*. Infantis is strongly associated with the spread of isolates harboring a large conjugative plasmid which confers resistance, virulence and fitness properties. This plasmid, termed pESI (plasmid of Emerging *S*. Infantis), was first reported in a multidrug-resistant (MDR) clone of *S*. Infantis detected since 2006 in Israel [7,8]. pESI was identified as a chimeric megaplasmid of ca. 280 kb, found to contain the origin of replication (*oriV*) and incompatibility region of IncP-1α plasmids, and the IncI leading and transfer region. Later on, pESI-like plasmids were also detected in *S*. Infantis from many other countries, including Italy, Switzerland, Denmark, England and Wales, the USA and South America, and they were shown to contain the IncFIB replicon [2,9,10,11,12,13,14,15]. All pSE1-like plasmids confer resistance against multiple antimicrobial agents, including clinically relevant antibiotics such as third-generation cephalosporins. So far, two distinct types of pESI-like plasmids bearing genes encoding extended spectrum β-lactamases (ESBL) have been reported: *bla*_CTX-M-1_-positive and *bla*_CTX-M-65_-positive plasmids, found in European and American isolates, respectively [16].

The molecular epidemiology of *S*. Infantis and its pESI-like plasmids in Europe has been investigated by whole-genome sequencing and bioinformatics analysis, using isolates from America as an outgroup [16]. This study revealed the heterogeneity of the European *S*. Infantis population, which is composed by multiple clusters defined at the core-genome level, confirming previous proposals of the polyphyletic nature of this serovar [11,17]. However, pESI-like variants, which were present in 64.1% of the isolates analyzed (a total of 362; [16]), were genetically more homogeneous and were distributed among different clonal lineages in most countries. It has been proposed that once acquired by an *S*. Infantis isolate, the conjugative megaplasmid would rapidly become established in the local population because of the selective advantages it confers to the host bacteria [16]. Such advantages are not only provided by the presence of antibiotics resistance genes, but also by genes coding for resistances to quaternary ammonium compounds and heavy metals, which are likewise widespread in other successful clones of non-typhoid serovars of *S. enterica* [13,18,19,20,21]. Moreover, pESI-like megaplasmids also provide genes involved in colonization, virulence and fitness, including those encoding two types of fimbriae (Fim and K88-like) and the siderophore yersiniabactin, as well as toxin–antitoxin systems (like CcdBA, PemKI) [7,16].

In a screen for *S. enterica* resistant to third-generation cephalosporins in Spanish hospitals, two isolates of *S*. Infantis resistant to cefotaxime were identified. The present study aimed to combine epidemiological information with experimental research and whole-genome sequencing (WGS) analysis for in-depth characterization of these isolates. This provided evidence for the migration of the *S*. Infantis clone carrying pESI-like plasmids of the “American type” from Peru to Spain.

## 2. Results

### 2.1. Origin of the Isolates, Genome Sequencing and Resistance Phenotypes

The two isolates, HUMV 13-6278 and HUMV 15-5476, were recovered in 2013 and 2015 at the “Hospital Universitario Marqués de Valdecilla” (HUMV) from fecal samples of small children with gastroenteritis (Table 1). The children did not require hospitalization and were attended to at Primary Care Centers associated with the hospital. In both cases, the onset of the disease coincided with their return to Spain after a trip to Peru.

As shown in Table 1 and Table 2, the latter compiling MIC values for relevant compounds, the two isolates were resistant to 14 out of the 21 antimicrobials tested, including cefotaxime (MIC of 32 µg/mL; used as the bases for selection of the isolates), nalidixic acid (MIC of 128 and >256 µg/mL for HUMV 13-6278 and HUMV 15-5476, respectively), pefloxacin (selected by EUCAST as a surrogate of ciprofloxacin to disclose clinical resistance to fluoroquinolones; https://eucast.org/clinical_breakpoints; accessed on 8 April 2022), ciprofloxacin (MIC of 0.125 µg/mL, resistant according to EUCAST criteria), and nitrofurantoin (MIC of 512 µg/mL). HUMV 15-5476 but not HUMV 13-6278 was also resistant to fosfomycin, displaying MIC values of 512 and 0.19 µg/mL, respectively. With regard to heavy metals, the two *S.* Infantis isolates were resistant to mercury as well as to inorganic and organic arsenic compounds (Table 2).

The draft genomes of HUMV 13-6278 and HUMV 15-5476 contained a total of 88 and 86 contigs, respectively, with an assembly size of about 4700 kb (Table 1). They shared the antigenic formula 7:r:1,5 and the sequence type ST32, both associated with *S*. Infantis. Each isolate harbored a single pESI-like plasmid of more than 310 kb (Table 1), termed pHUMV 13-6278 and pHUMV 15-5476, respectively. The two plasmids carried the IncFIB replicon, the IncP origin of replication (*oriV*) and a gene for a MOBP-family relaxase. Additionally, plasmid typing identified the *ardA_2*, *pilL_3*, *sogS_9*, *trbA_21* alleles of the pMLST scheme designed for the IncI1 incompatibility group, but lacked the *repI1* gene. Based on the detected alleles, the closest related sequence type within the IncI pMLST scheme was ST71.

### 2.2. Genetic Bases of Antimicrobial Drug Resistance

The resistance genes detected in the genomes of the two isolates by bioinformatics analysis were in agreement with the observed phenotypes (Table 1). Both isolates contained *bla*_CTX-M-65_ (ampicillin and cefotaxime), *floR* (chloramphenicol), *aac(3)-IVa*, *aph(3′)-Ia*, *aph(4)-Ia*, *aadA1* (aminoglycosides), *tet*(A) (tetracycline), *sul1* and *dfrA14* (sulfonamides and trimethoprim, respectively), together with genes conferring resistance to heavy metals: locus *merRTPCADE* (mercury), *arsA* and *arsD* (inorganic arsenic compounds), *arsH* (organic arsenicals), *arsR2* (metal-responsive transcriptional regulator), and *qacEΔ1* (although resistance to disinfectants was not experimentally tested). In the case of HUMV 15-5476, fosfomycin resistance was mediated by *fosA3*. All these genes were located on the pESI-like plasmids, as part of two complex resistance regions: R1 and R2, comprising approximately 36 or 32 kb (depending on the presence or absence of *fosA3*) and 21 kb, respectively. R1 and R2 were separated by DNA of the pESI-like backbone, and contained a wealth of transposable elements belonging to multiple families, including several copies of IS26 (Figure 1).

The resistance regions of the two plasmids differed only by the presence of the IS*26*-*fosA3*-IS*26* pseudo-compound transposon in R1 of pHUMV 15-5476, which was lacking in the corresponding region of pHUMV 13-6278, and by a copy of IS26 located at the 5′-end of R2 in the former but not in the latter. Consistent with the high conservation of the pESI-like plasmids [16], regions identical to those of the plasmids under study were found in other plasmids of this group. However, there was also considerable variation, as shown by their comparison with some key examples (Figure 1), which include the original pESI plasmid, detected in a human isolate from Israel (119944), which lacks most of the R1 region [7,8]. A higher degree of similarity was found with two *bla*_CTX-M-65_-positive plasmids, with or without *fosA3* (FSIS1502169 and CVM N18S2198, respectively; [14,24]), identified in isolates from chicken samples in the USA.

Apart from the plasmid-encoded genes, the two isolates displayed a point mutation in the chromosomal *gyrA* gene (G259T), resulting in D87Y exchange in the protein, which could account for nalidixic acid resistance. In addition, both strains also carried a single-nucleotide polymorphism in their chromosomal *parC* gene (C170G), leading to T57S substitution in the protein. Finally, nitrofurantoin resistance of the two isolates was associated with premature stop codons in the *nsfA* and *nsfB* genes, producing truncated forms of the NsfA and NsfB nitroreductases, with only 158 and 136 amino acids, as opposed to 240 and 217 amino acids in the wild type, respectively.

### 2.3. Phylogenetic Analysis

Figure 2 shows the phylogenetic position of the two HUMV isolates in the context of other *S*. Infantis isolates. Since *S.* Infantis is a broiler-associated pathogen, we chose a selective comparison of sequences of isolates from humans, chickens or the chicken farm environment in different countries of Europe and America, and in Israel (Appendix A). All isolates harbored a pESI-like plasmid, which carried the *bla*_CTX-M-1_, the *bla*_CTX-M-65_ gene, or neither. The Spanish isolates, which differed by only 12 SNPs, belonged to a clonal lineage that groups all *bla*_CTX-M-65_-positive isolates together with isolates lacking a *bla*_CTX-M_ gene. The minimum and maximum number of SNPs between isolates of this cluster ranged from 3 to 60, and the distance with respect to the 119944 strain from Israel, which contains the original pESI plasmid, ranged from 93 to 127 SNPs.

## 3. Discussion

The present study examined two clinical isolates of *S*. Infantis harboring the *bla*_CTX-M-65_ gene on pESI-like megaplasmids. As other *bla*_CTX-M_ genes, it confers resistance to most β-lactam antibiotics, including penicillins, broad spectrum cephalosporins and monobactams, but not to amoxicillin-clavulanic acid, cephamycins and carbapenems [25]. As such, it conveys resistance to third-generation cephalosporins, which currently are first-line drugs for the management of severe *S. enterica* infections. Thus, the presence of *bla*_CTX-M-65_ is a cause of concern, particularly when associated with resistance to other classes of medically relevant antimicrobials. In fact, the two isolates were resistant to antimicrobials belonging to eight classes: penicillins, cephalosporins, phenicols, aminoglycosides, tetracyclines, inhibitors of folate synthesis, (fluoro)quinolones, and nitrofurans, as confirmed by disk diffusion assays complemented with MIC determinations for selected compounds. HUMV 15-5476, but not HUMV 13-6278, was additionally resistant to fosfomycin.

Most of the resistance genes found were located on pESI-like megaplasmids, but resistances to (fluoro)quinolones and nitrofurantoin were mediated by chromosomal loci. Thus, the two isolates had single-point mutations both in *gyrA* (G259T→D87Y) and *parC* (C170G→T57S). The former mutation, which is now widespread among *S*. Infantis, was originally identified as responsible for nalidixic acid resistance in isolates from Israel. This trait was already present in a limited number of pre-emergent isolates, which were obtained before 2006 and lacked pESI, while becoming fixed in the *gyrA* gene of the emergent pESI-positive clone (recorded after 2006), adding to its MDR phenotype [3,7]. In addition to D87Y, three other point mutations in *gyrA* (D87G, S83Y and S83F) were later associated with (fluoro)quinolone resistance in an extensive collection of *S*. Infantis recovered in Europe from animals, meat, feed and humans [16]. In that work, a single-(fluoro)quinolone-resistant isolate tested negative for known mutations in *gyrA*, but presented the T57S substitution in the ParC protein. The latter change has been previously found both in resistant and in susceptible strains of different serovars of *S. enterica*, with or without concomitant mutations in *gyrA* [16,26,27,28,29]. The contribution of the T57S substitution to (fluoro)quinolone resistance in the isolates thus remains unclear. PMQR (Plasmid-Mediated Quinolone Resistance) mechanisms, which confer low level, albeit clinically relevant, resistance to fluoroquinolones, were not present in our isolates but have already been reported in *S*. Infantis from different countries [16].

In *S. enterica*, as well as in *Escherichia coli*, resistance to nitrofurans has been associated with alterations affecting the *nfsA* and *nfsB* genes that encode type I (i.e., oxygen insensitive) nitroreductases. Amongst other substrates, these enzymes appear to reduce nitrofurans, generating active intermediates with antimicrobial activity. Nitrofurans are used for treating uncomplicated urinary tract infections caused by a wide spectrum of Gram-positive and Gram-negative bacteria [30,31]. In both *E. coli* and *S*. Typhimurium, a stepwise increase in resistance to nitrofurantoin has been correlated with sequential inactivation first of *nfsA* and then of *nfsB*, with alterations in the two genes being required to achieve full resistance [22,32,33]. The *S*. Infantis isolates in the present study were highly resistant to nitrofurantoin (Table 2). In agreement with this, both carried premature stop codons in the two *nfs* genes, leading to truncated forms of the proteins. The same mutation in *nfsA* was previously detected in *S*. Infantis from Israel, without reference to alterations in the *nfsB* gene [7]. Yet, sequence comparisons of the *nsfB* gene from our isolates and strain 119944, obtained in 2008 from a clinical sample in Israel [3], revealed the presence of the mutation (data not shown). Accordingly, the isolate from Israel was likely to display high-level resistance to nitrofurantoin, although the MIC of nitrofurantoin was not reported.

As previously shown for other *S*. Infantis isolates, the pESI-like megaplasmids played a major role in the MDR phenotype of the isolates under study. They not only harbor numerous antibiotic resistance genes, but also genes for resistance to mercury (*merRTPCADE* locus) and arsenic (*arsA*, *arsD*, *arsH* and *arsR2*) compounds. In the case of arsenic, toxicity depends on the chemical form (inorganic or organic) and oxidation state (trivalent or pentavalent species) of the compound, with As(III) arsenite being more toxic than As(V) arsenate. The product of *arsA* is an ATPase that interacts with ArsB to form an arsenite efflux pump energized by ATP hydrolysis. It is of note that *arsA*, but not *arsB*, is present in the pESI-like plasmids of the isolates under study, which were nevertheless resistant to NaAsO_2_ [As(III)]. However, besides canonical *ars* genes, additional genes have been discovered that broaden the spectrum of arsenic tolerance [34]. Apart from interacting with ArsB, the ArsA ATPase has been proposed to form primary arsenite transporters in association with different membrane proteins [35]. The most likely candidate would be Acr3, but the *acr3* gene was not found in the genome of our isolates. The product of *arsD*, a gene also present in pESI-like plasmids, acts as a metallochaperone that binds arsenite and transfers it to the ArsA ATPase for export [36]. Finally, the ArsH protein, which is under the control of the ArsR2 transcriptional regulator, encodes an organoarsenical oxidase that transforms trivalent methylated and aromatic arsenicals into their less toxic pentavalent species [37]. Accordingly, the two isolates described herein were resistant to the aromatic As(III) phenylarsine oxide. Of note, although *mer* and *ars* genes are frequently found in pESI-like plasmids, their association with phenotypic resistance of *S*. Infantis has not been previously tested [38]. The selective pressure caused by the use of heavy metals, and also of nalidixic acid, tetracyclines, sulfonamides and nitrofurans in livestock production, could have contributed to the selection of pESI-positive isolates of *S*. Infantis, as it was probably the case for other successful clones of different non-typhoidal serovars of *S. enterica* [13,18,19,20,21].

Of the multiple resistance genes harbored by the pESI-like plasmids under study, only *fosA3* was differentially present in HUMV 15-5476, which was highly resistant to fosfomycin (Table 2). This broad-spectrum bacteriolytic antibiotic was originally approved for treating uncomplicated urinary tract infections in the early 1970s. However, the major threat posed by antimicrobial resistance to human health, and the shortage of new antimicrobial agents, has led to re-evaluating older antibiotics, including fosfomycin, for the treatment of infections caused by multidrug-resistant bacteria [39]. Therefore, the increasing rate of fosfomycin resistance, which is mainly mediated by inactivating enzymes, is a cause of concern. Like other *fosA* genes, the *fosA3* gene of the pESI-like plasmid of HUMV 15-5476 encodes a glutathione-S-transferase, which catalyzes the opening of the epoxide ring of fosfomycin by adding glutathione [40]. In *S*. Infantis, *fosA3* has been found in the subgroup of pESI-like plasmids positive for *bla*_CTX-M-65_, but not in those carrying *bla*_CTX-M-1_ [16].

It is interesting to note that, as previously reported for other pESI-like plasmids, those of the HUMV isolates not only contained antimicrobial-resistant genes, but also genes associated with virulence, colonization and fitness, including genes for fimbriae, for the siderophore yersiniabactin, and for toxin-antitoxin systems which are involved in the post-segregational killing of bacteria that lose the plasmid. Taken together, these determinants confer selective advantages which explain the epidemiological success of the isolates that have managed to acquire this self-transmissible plasmid [7,41].

It is finally of note that isolates harboring pESI-like plasmids with *bla*_CTX-M-65_ ± *fosA3*, like those reported herein, form a well-defined cluster within the phylogenetic tree of *S*. Infantis ([13,16]; Figure 2). They have been reported in humans and along the poultry production chain in the USA and South America, with only a limited number of isolates being detected in other livestock animals [14,24,42,43,44,45,46]. Isolates carrying pESI-like plasmids positive for *bla*_CTX-M-65_ ± *fosA3* have also been associated with human clinical cases in Europe, specifically in patients with a travel history to America [10,12,16]. However, to the best of our knowledge, this is the first report of such isolates in Spain, with a clearly established link to Peru. In Spain, information is also lacking on the possible presence of *S*. Infantis carrying pESI-like plasmids of the “European type”, positive for *bla*_CTX-M-1_ instead of *bla*_CTX-M-65_, which are circulating in food-animal production systems of many other European countries [16]. Within the One Health Concept, promoted by the World Health Organization, further studies are required to investigate the occurrence of such isolates in Spanish livestock, and particularly along the poultry food chain. WGS analyses could certainly assist in the detection and detailed characterization of isolates belonging to the different MDR CTX-M-producing lineages of *S*. Infantis, and to establish their phylogenetic relationships with those from other countries.

## 4. Materials and Methods

### 4.1. Bacterial Isolates and Antimicrobial Susceptibility Testing

The two cefotaxime-resistant isolates of *S*. Infantis, HUMV 13-6278 and HUMV 15-5476, selected for the present study, were detected at the “Hospital Universitario Marqués de Valdecilla” (HUMV), Santander, Cantabria, Spain (Table 1). They were obtained from fecal samples of small children with gastroenteritis, using selective culture media (Selenite broth and Hecktoen agar; bioMerieux, Marcy l’Etoile, France). Identification was performed with MALDI-TOF MS, following the manufacturer’s instructions (Bruker Daltonics, Billerica, MA, USA).

Susceptibility to antimicrobial agents was determined by automated MicroScan NC 53 (Beckman Coulter, CA, USA), complemented with disk diffusion assays. For the latter, the following compounds (Oxoid, Madrid, Spain, except for pefloxacin purchased from Bio-Rad, Alcobendas, Madrid, Spain) were used, with the amount per disk in µg shown in parenthesis: ampicillin (10), amoxicillin-clavulanic acid (30), cefepime (30), cefotaxime (30), cefoxitin (30), erthapenem (10), chloramphenicol (30), amikacin (30), gentamicin (10), kanamycin (30), streptomycin (10), tobramycin (10), azithromycin (15), nalidixic acid (30), ciprofloxacin (5), pefloxacin (5), sulfonamides (300), tetracycline (30), trimethoprim (5), fosfomycin (300) and nitrofurantoin (300). The minimum inhibitory concentrations (MIC) of cefotaxime, ciprofloxacin and fosfomycin were determined by E-test (bioMérieux, Marcy l’Étoile, France), and the MIC of nitrofurantoin by a broth microdilution assay with concentrations ranging from 4 to 512 µg/mL. Results were interpreted according to EUCAST (European Committee on Antimicrobial Susceptibility Testing) guidelines (https://eucast.org/clinical_breakpoints/; accessed on 8 April 2022) or to CLSI [47], in the case of nalidixic acid which is not contemplated by EUCAST. To obtain additional information on the resistance properties of the *S*. Infantis isolates, the MIC of heavy metals were also determined by broth microdilution assays, using HgCl_2_ (0–256 µg/mL) and the following inorganic and organic arsenic compounds (Sigma-Aldrich, Merck Life Science, Madrid, Spain): NaAsO_2_ (0–256 µg/mL), Na_2_HAsO_4_∙7H_2_O (0–256 µg/mL) and phenylarsine oxide (0–16 µg/mL). *S*. Typhimurium LT2, *S*. Typhimurium LSP 146/02 and *S*. Typhimurium 4,5,12:i:- (monophasic) LSP 389/97 were included as negative or positive controls [21,22,23].

### 4.2. Whole-Genome Sequencing and Bioinformatics Analysis

For WGS, the genomic DNA of the isolates was purified from overnight cultures grown in Luria-Bertani (LB) broth, using the GenEluteTM Bacterial Genomic DNA Kit, according to the manufacturer’s protocol (Sigma-Aldrich). WGS was performed with Illumina technology, at the “Centro de Investigación Biomédica”, La Rioja (CIBIR), Spain. Paired-end reads of 100 nt, obtained from PCR-free fragment libraries of ca. 500 bp, were assembled de novo using the VelvetOptimiser.pl script implemented in the on line version of PLACNETw (https://omictools.com/placnet-tool; accessed on 6 June 2019). This tool also allows distinguishing contigs of chromosomal or plasmid origin [48]. The assembled genomes were deposited in GenBank under the accession numbers included in the “Data Availability Statement” (see below), and annotated by the NCBI Prokaryotic Genome Annotation Pipeline (PGAP; https://www.ncbi.nlm.nih.gov/genome/annotation_prok/; accessed on 4 August 2021). Relevant parameters regarding the quality of the assemblies are shown in Appendix A. Bioinformatics analysis was performed with PLACNETw, MOBscan (a web application for the identification of relaxase MOB families [49]), and with different tools available at the Center for Genomic Epidemiology (CGE) of the Technical University of Denmark (DTU) (https://cge.cbs.dtu.dk/services/; accessed on 7 June 2020). These included SeqSero 1.2 [50], MLST 2.0 [51], ResFinder 4.1 [52,53], PlasmidFinder 2.1 and pMLST 2.0 [54].

Once identified by PLACNETw, plasmid contigs carrying resistance genes were joined by PCR amplification (using the primers shown in Appendix A), followed by Sanger sequencing of the obtained amplicons, when required. The annotation of the assembled resistance regions were manually curated with the aid of blastn, blastp (https://blast.ncbi.nlm.nih.gov/Blast.cgi; accessed on 28 April 2022) and CLONE Manager Professional (CmSuit9). The genetic organization of these regions was represented with EasyFig BLASTn (https://mjsull.github.io/Easyfig/; accessed on 7 May 2022).

### 4.3. Phylogenetic Analysis

A phylogenetic tree based on single-nucleotide polymorphisms (SNPs) was built with the CSI phylogeny 1.4 tool (https://cge.cbs.dtu.dk/services/CSIPhylogeny/; accessed on 6 May 2022 [55]). The pipeline was run with default parameters, setting a bootstrap replication value of 1,000 to generate the consensus tree [56]. Apart from the genomes of the HUMV isolates, 42 other *S*. Infantis genomes retrieved from databases were included in the analysis (see Appendix A for accession numbers and additional details). The genome of isolate 119944 from Israel (accession No. GCA_000506925.1) was used as the reference for SNP calling, and the resulting SNP matrix is shown in Appendix A.

## 5. Conclusions

This study documents the import of the *S*. Infantis MDR clone which carries the *bla*_CTX-M-65_ within pESI-like megaplasmids from Peru into Spain. To the best of our knowledge, this is the first report on the presence of this clone in Spain, which has probably emerged in South America [46]. We also first experimentally demonstrated the resistance of the clone to mercury and to inorganic and organic arsenicals. Further studies will be required to establish the actual epidemiology of these and other CTX-M-producing *S*. Infantis lineages in Spain.

## Figures and Tables

**Figure 1 antibiotics-11-00786-f001:**
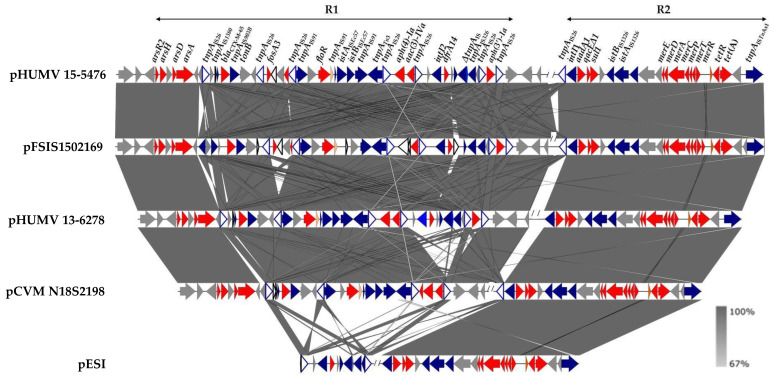
Comparison between the assembled resistance regions (R1 and R2) of the pESI-like plasmids pHUMV 13-6278 and pHUMV 15-5476 found in the *Salmonella enterica* serovar Infantis isolates under study with related regions from other pESI-like plasmids (accession No. CP016407, CP082522 and GCA_000506925.1 for plasmids of the FSIS1502169, CVM N18S2198 isolates and plasmid pES1 of the 119944 isolate, respectively). Coding regions are represented by arrows indicating the direction of transcription and colored according to their function: red, resistance; blue, insertion sequences, with the multiple copies of IS26 highlighted by arrowheads with white background and blue border lines; grey, other roles; white with black border lines, hypothetical proteins. The alignments were created with EasyFig BLASTn. The gray shading between regions reflects nucleotide sequence identities according to the scale shown at the right lower corner of the figure.

**Figure 2 antibiotics-11-00786-f002:**
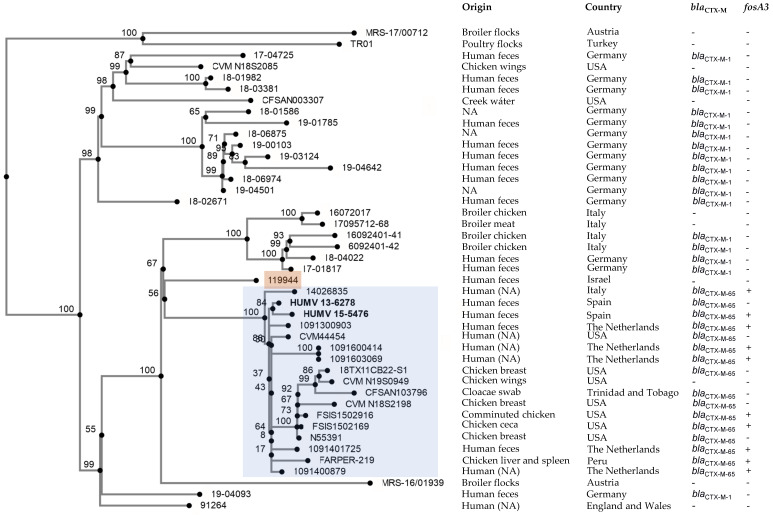
Phylogenetic position of *Salmonella enterica* serovar Infantis HUMV 13-6278 and HUMV 15-5476 from a Spanish hospital, in the context of *S*. Infantis isolates from different sources and countries. The tree was constructed with the CSI Phylogeny 1.4 (https://cge.cbs.dtu.dk/services/CSIPhylogeny/, accessed on 6 May 2022) using the genome of *S*. Infantis 119944 as the reference for SNP calling. Values at each node represent percent bootstrap support based on 1000 replicates. The cluster containing the Spanish isolates (shown in bold) is highlighted in blue, and the 119944 isolate is emphasized in red. Relevant information related to the isolates is shown at the right of the figure. Accession numbers of the genomes and the pairwise distance matrix used to construct the phylogenetic tree are shown in Appendix A, respectively.

**Table 1 antibiotics-11-00786-t001:** Origin and properties of clinical cefotaxime-resistant isolates of *Salmonella enterica* serovar Infantis ST32.

HUMV Isolate ^a^	Patient Sex ^b^/Age	Travel History	Clinical Sample	Resistance Pattern ^c^ Antibiotic Resistance Genes ^d^	Chromosome (Size bp) Plasmid ^e^ (Size bp)
13-6278	M/23 mth	Peru	Feces	[NAL, CIP, PEF], NIT	
				[*gyrA* D87Y, *parC* T57S], [Δ*nfsA*, Δ*nfsB*]	Chromosome (4,686,236)
				[AMP, CTX], CHL, [GEN, TOB, KAN, STR], TET, [SUL, TMP]	
				*bla*_CTX-M-65_, *floR*, [*aac(3)-IVa*, *aph(3′)-Ia*, *aph(4)-Ia*, *aadA1*], *tet*(A), [*sul1*, *dfrA14*]	IncFIB (313,645)
15-5476	M/12 mth	Peru	Feces	[NAL, CIP, PEF], NIT	
				[*gyrA* D87Y, *parC* T57S], [Δ*nfsA,* Δ*nfsB*]	Chromosome (4,682,901)
				[AMP, CTX], CHL [GEN, TOB, KAN, STR], TET, [SUL, TMP], FOS	
				*bla*_CTX-M-65_, *floR*, [*aac(3)-IVa*, *aph(3′)-Ia*, *aph(4)-Ia*, *aadA1*], *tet*(A), [*sul1*, *dfrA14*], *fosA3*	IncFIB (317,684)

^a^, HUMV, “Hospital Universitario Marqués de Valdecilla”, Santander, Cantabria, Spain. ^b^, M, male; mth, months. ^c^, NAL, nalidixic acid; CIP, ciprofloxacin; PEF, pefloxacin; NIT, nitrofurantoin; AMP, ampicillin; CTX, cefotaxime; CHL, chloramphenicol; GEN, gentamycin; TOB, tobramycin; KAN, kanamycin; STR, streptomycin; TET, tetracycline; SUL, sulfonamides; TMP, trimethoprim; FOS: fosfomycin (note that resistance to hygromycin, which could be conferred by the *aph(4)-1a* gene, was not experimentally tested). Antimicrobials belonging to the same class are combined in brackets. ^d^, Genes conferring resistance to antimicrobials of the same class are combined in brackets. ^e^, Inc, plasmid incompatibility group. The IncFIB megaplasmids of the isolates under study were named pHUMV 13-6278 and pHUMV 15-5476.

**Table 2 antibiotics-11-00786-t002:** Minimum inhibitory concentration (MIC, given in µg/mL) of relevant antibiotics and heavy metals determined for *Samonella enterica* serovar Infantis strains from a Spanish hospital.

Strain ^a^	CTX ^b^	NAL ^b^	CIP ^b^	NIT ^b^	FOS ^b^	HgCl_2_	NaAsO_2_	Na_2_HAsO_4_	Phenylarsine Oxide
HUMV 13-6278	32	128	0.125	512	0.19	64	128	2048	8
HUMV 15-5476	32	>256	0.125	512	512	32	128	512	8
LT2	4	3	0.016	32	0.019	4	64	128	0.25
LSP 146/02	nd	nd	nd	256	nd	32	32	256	0.5
LSP 389/97	nd	nd	nd	nd	nd	32	64	128	4

^a^, HUMV, “Hospital Universitario Marqués de Valdecilla”, Santander, Cantabria, Spain; LSP, “Laboratorio de Salud Pública”, Asturias, Spain. *S. enterica* serovar Typhimurium strain LT2 was included as negative control. *S. enterica* serovar Typhimurium strain LSP 146/02 and *S. enterica* serovar 4,5,12:i:- strain LSP 389/97 were included as positive controls of resistance to some of the antimicrobials tested [21,22,23]. ^b^, CTX, cefotaxime; NAL, nalidixic acid; CIP, ciprofloxacin; NIT, nitrofurantoin; FOS: fosfomycin.

## Data Availability

The genome sequences generated in the present study were deposited in GenBank under accession numbers JAICDN000000000 and JAICDM000000000 for *Salmonella enterica* serovar Infantis HUMV 13-6278 and HUMV 15-5476, respectively.

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
