# Peer review of "Genomic Analysis of Two MDR Isolates of Salmonella enterica Serovar Infantis from a Spanish Hospital Bearing the blaCTX-M-65 Gene with or without fosA3 in pESI-like Plasmids"

_antibiotics, 2022, doi:10.3390/antibiotics11060786_

Round 1

Reviewer 1 Report

The manuscript by Vázquez and colleagues characterises in detail two strains of S. Infantis isolated in Spanish hospitals. S. Infantis has been a topic of multiple recent publications, partially due to its rapid spread throughout broiler populations, but also because of the acquisition of blaCTX-M genes leading to expression of ESBL. It is interesting that the two strains described by the authors harbour blaCTX-M-65 gene and not the 'European variant' M-1. Unfortunately, there are no current publications that describe the occurrence of ESBL producing strains and their respective antimicrobial resistance genes in the Spanish livestock.

The article is written in a detailed and clear manner, and the authors used modern bioinformatics tools to describe the genomic features of the two ESBL strains. I have only minor comments which should be addressed before the manuscript can be accepted.

Introduction - please add a paragraph about the metal and biocide resistance in Salmonella (and S. Infantis in particular)

- please add a clear aim of the study, how can results of the study be applied or help to further the research

Discussion

- please discuss the results in the context of Salmonella Infantis epidemiology in Spain

- please discuss the results in the context of One Health concept and the prevalence of S. Infantis in broiler in Spain

Author Response

Dear Reviewer 1,

Please find enclosed our point by point answers to your comments and requirements, which are shown below each of them, preceded by three asterisks. We would like to thank you for your efforts and comments, which have really helped to improve the manuscript.

Reviewer 1

Comments and Suggestions for Authors

The manuscript by Vázquez and colleagues characterises in detail two strains of S. Infantis isolated in Spanish hospitals. S. Infantis has been a topic of multiple recent publications, partially due to its rapid spread throughout broiler populations, but also because of the acquisition of blaCTX-M genes leading to expression of ESBL. It is interesting that the two strains described by the authors harbour blaCTX-M-65 gene and not the 'European variant' M-1. Unfortunately, there are no current publications that describe the occurrence of ESBL producing strains and their respective antimicrobial resistance genes in the Spanish livestock.

The article is written in a detailed and clear manner, and the authors used modern bioinformatics tools to describe the genomic features of the two ESBL strains. I have only minor comments which should be addressed before the manuscript can be accepted.

***Thank you very much.

Introduction - please add a paragraph about the metal and biocide resistance in Salmonella (and S. Infantis in particular)

***The presence of heavy metal and biocide resistance genes in S. Infantis megaplasmids, and the relevance of these resistances not only for S. Infantis but also for other successful clones of Salmonella is now referred to, both in the Introduction and the Discussion sections (lines 78-81 and 250-275).

- please add a clear aim of the study, how can results of the study be applied or help to further the research.

***The aim of the study has been rephrased (lines 86-90). The way in which the obtained results can prompt future research has also been added (lines 384-386).

Discussion

- please discuss the results in the context of Salmonella Infantis epidemiology in Spain

- please discuss the results in the context of One Health concept and the prevalence of S. Infantis in broiler in Spain

***Done as required, placing the focus on CTX-M producers (lines 302-312). Please note that Dr. J. Rodriguez-Lozano (added as a new author) was able to contact the families of the children who suffered the S. Infantis infections. The fact that both developed the symptoms after their return from a trip to Peru has been included into the manuscript.

Reviewer 2 Report

Dear authors, I have read your manuscript with great interest and I must congratulate you for your work. Attached some suggestions for improvement.

INTRODUCTION

Correctly written, direct to the research topic and placed in the context of the paper.

All abbreviations and acronyms must be described before. What does the acronym UE mean?

RESULTS

Avoid explaining results and repeating what has already been shown in the tables.

DISCUSSION

Consider the need for molecular and automated methods for the best diagnosis and treatment of patients affected by these germs. I suggest using this reference: Batista-Araújo MR, Ferreira-Seabra L, Oliveira-Sant'Anna L, Sanches-dos-Santos L. Identification of Salmonella enterica serovar Typhi strain from a young Brazilian patient: the relevance of automated microbiological methods for the rapid diagnosis of systemic infections Microbes Infect. Chemother. 2022; 2: e1295 https://doi.org/10.54034/mic.e1295

Insist on the transmission of resistance genes to this type of germs, and even to other bacteria.

MATERIALS AND METHODS

adequately described

CONCLUSIONS

I think the entire manuscript is relevant and interesting, however, here I should indicate that it is a report.

Author Response

Dear Reviewer 2,

Please find enclosed our point by point answers to your comments and requirements, shown below each of them, preceded by three asterisks. We would like to thank you for your time and suggestions that have helped us to improve the manuscript.

Comments and Suggestions for Authors

Dear authors, I have read your manuscript with great interest and I must congratulate you for your work. Attached some suggestions for improvement.

***Thank you.

INTRODUCTION

Correctly written, direct to the research topic and placed in the context of the paper.

All abbreviations and acronyms must be described before. What does the acronym UE mean?

***Sorry, UE was a typo (abbreviation for European Union in Spanish). It has been corrected in the revised version (line 51).

RESULTS

Avoid explaining results and repeating what has already been shown in the tables.

***Most data shown in the tables have been eliminated from the text.

DISCUSSION

Consider the need for molecular and automated methods for the best diagnosis and treatment of patients affected by these germs. I suggest using this reference: Batista-Araújo MR, Ferreira-Seabra L, Oliveira-Sant'Anna L, Sanches-dos-Santos L. Identification of Salmonella enterica serovar Typhi strain from a young Brazilian patient: the relevance of automated microbiological methods for the rapid diagnosis of systemic infections Microbes Infect. Chemother. 2022; 2: e1295 https://doi.org/10.54034/mic.e1295

Insist on the transmission of resistance genes to this type of germs, and even to other bacteria.

***The manuscript by Batista-Araújo et al. demonstrates the efficacy of MALDI-TOF MS for rapid and accurate identification of pathogens, which is essential for diagnosis. In most Spanish hospitals, including tertiary hospitals like the HUMV in the present study, MALDI-TOF-MS is routinely used for bacterial identification. This was in fact the case for the S. enterica isolates in the present study, as indicated in Materials and Methods. However, the paper does not deal with MALDI-TOF MS, but with Whole Genome Sequencing and its relevance for accurate bacterial identification and typing, as well as for detection of plasmids and antimicrobial resistance genes. This really allows identification of the precise clones of pathogenic bacteria which are circulating in a certain geographical region, and to place them in the context of the world-wide infection history. The usefulness of the WGS approach for this aim has been added to the discussion (lines 309-312).

***The facts that the pESI-like plasmids are conjugative and that they are playing an important role on transmission, not only of resistance genes but also of colonization, virulence and fitness traits, have been emphasized (lines 289-295).

MATERIALS AND METHODS

adequately described

***Thank you.

CONCLUSIONS

I think the entire manuscript is relevant and interesting, however, here I should indicate that it is a report.

***The conclusions have been thoroughly modified to adapt them to the changes made in the manuscript (lines 380-386).
